# Targeting c-MET Alterations in Cancer: A Review of Genetic Drivers and Therapeutic Implications

**DOI:** 10.3390/cancers17091493

**Published:** 2025-04-29

**Authors:** Michelle Ji, Shridar Ganesan, Bing Xia, Yanying Huo

**Affiliations:** 1Department of Radiation Oncology, Rutgers Cancer Institute, 195 Little Albany Street, New Brunswick, NJ 08903, USA; michelle.ji@mail.utoronto.ca (M.J.); xiabi@cinj.rutgers.edu (B.X.); 2University of Toronto, Toronto, ON M5S 1A1, Canada; 3NYU Langone Perlmutter Cancer Center, 160 East 34th Street, New York, NY 10016, USA

**Keywords:** *MET*, c-MET, *MET* exon 14 deletion, *MET* gene fusions, MET-driven cancers, *MET* mutations, targeted therapies, oncogenesis, drug development

## Abstract

This review focuses on the role of the *MET* proto-oncogene in cancer and how alterations in this gene contribute to tumor growth and spread. Understanding how its product, c-MET, functions and the ways it can become overactive in cancer cells is essential for developing new targeted treatments. By summarizing the current knowledge on c-MET-related dysregulations in various cancers, we aim to provide a clear picture of c-MET as a target for cancer therapies. We discuss the types of drugs currently approved or in development that aim to block c-MET signaling. The goal is to help researchers and clinicians better understand c-MET’s role in cancer and guide future studies on targeted therapies, ultimately aiming to improve patient outcomes.

## 1. c-MET Protein Structure

*MET* is a proto-oncogene that encodes c-MET, also known as hepatocyte growth factor receptor (HGFR) [1,2], a member of the receptor tyrosine kinase family. c-MET is formed by an extracellular alpha chain and a transmembrane beta chain that consists of the semaphorin (SEMA), PSI (plexins, semaphorins, and integrins), immunoglobulin-like (IPT), transmembrane (TM), juxtamembrane (JM), catalytic, and carboxyl terminal docking domains [3,4,5]. Disulfide bonds link the alpha and beta chains. The structure of c-MET is depicted by Figure 1. HGF, or hepatocyte growth factor, is translated as an inactive, single-chain precursor ligand that is proteolytically converted into an active heterodimer upon binding to c-MET [5]. Ligand-receptor binding induces dimerization of c-MET via the SEMA domain, which folds into a seven-bladed β-propeller structure and is a necessary component for HGF to bind and activate the c-MET receptor [6,7]. The PSI domain’s function is still relatively unknown, although recent studies have shown that it plays a role in c-MET protein maturation and may serve as a chaperone for other protein folding [8]. The IPT domain also assists in HGF binding; its immunoglobulin-like regions are necessary for the NK1 domain of HGF to bind to c-MET [5,9]. The IPT and SEMA domains help c-MET bind to HGF, resulting in trans-phosphorylation on Y1234 and Y1235, part of the cytoplasmic domain that includes the JM domain, catalytic tyrosine kinase domain, and carboxyl-terminal tail that contains multiple tyrosine phosphorylation sites for signaling protein interaction. Before the cytoplasmic domain lies the TM domain, encoded by exon 13, which contains a hydrophobic region spanning the cell membrane.

Following the extracellular and transmembrane domains, the JM domain, encoded by exon 14, starts a series of intracellular domains of c-MET. The JM domain is a crucial region of c-MET, responsible for the negative regulation of the receptor and downstream signaling pathways. The JM domain has the direct binding site of CBL, an E3 ubiquitin ligase, which causes ubiquitination and degradation of the receptor [10]. The JM domain is also the site of multiple negative regulatory mechanisms. The first negative regulatory mechanism relies on the phosphorylation of serine 985. When serine 985 is phosphorylated, it inhibits the phosphorylation of the tyrosines required for further downstream signaling, such as Y1234, Y1235, Y1349, and Y1356 [11], thereby influencing how the cell responds to HGF. In a study by Nakayama et al., the authors used carbon tetrachloride (CCl_4_) to induce a non-lethal liver injury in mice that would require tissue and cell regeneration to restore normal liver function, one of the functions HGF/c-MET pathway is responsible for [12]. They found that liver regeneration was associated with decreased S985 phosphorylation levels, suggesting its negative regulatory relationship with activation of c-MET [12]. The second site of negative regulation in the JM domain is residue Y1003 [11]. After HGF activates c-MET receptor, Y1003 phosphorylation recruits CBL [13,14], which leads to c-MET ubiquitination, degrading c-MET when its functions are completed [11]. The JM domain also plays a key role in the activation of c-MET, as a study by Zhen et al. revealed that removing the first 39 amino acids of this domain resulted in a complete loss of activation potential and kinase activity. As a result, partial loss of the JM domain may inhibit kinase activation, whereas total loss of the domain leads to full constitutive, unregulated kinase activation. Additionally, when the 39 amino acids were replaced with the tetratricopeptide repeat sequence—a structural motif consisting of a degenerate 34-amino-acid tandem repeat sequence that facilitates protein–protein interactions—kinase activity was restored [15].

The trans-phosphorylation of Y1234 and Y1235 is followed by phosphorylation of Y1349 and Y1356 at the carboxyl-terminal tail [16]. Due to these phosphorylation events, further downstream signaling occurs, playing key roles in cellular survival, embryogenesis, cellular migration, and invasion. The carboxyl-terminal tail plays a crucial role in allowing downstream activity to occur. The phosphorylation of Y1349 and Y1356 recruits SH2 effectors, such as the GRB2/SOS complex, p85 complex, and Gab1, further activating downstream pathways [17]. In a study conducted by Bardelli et al., the researchers manipulated the carboxyl-terminal tail to understand its involvement with the rest of the c-MET receptor. Bardelli et al. investigated the role of tyrosine residues Y1349 and Y1356 in the C-terminal tail of the c-MET receptor by substituting them with non-phosphorylatable amino acids. This modification inhibited c-MET autophosphorylation and downstream signaling, highlighting the critical importance of these residues in receptor activation and signal transduction [17]. Potential explanations for this inhibition include ATP depletion due to peptide activity or direct inhibition caused by the peptide interacting with c-MET’s active site [17]. The role of the carboxyl-terminal tail in inhibiting autophosphorylation and downstream signaling highlights a potential pathway for future drug development, especially in treating patients with c-MET overexpression to prevent excessive activation of the protein and its signaling pathways.

## 2. c-MET Signaling Pathway

Following HGF binding, c-MET activates three main downstream pathways (Figure 2). When Y1349 and Y1356, are phosphorylated, they form a unique SH2 recognition motif crucial for interpreting and translating upstream signals into downstream cellular responses [14]. When the two tyrosine residues are mutated, the SH2 binding domain is unable to form properly, resulting in inappropriate signals to be relayed downstream. The first pathway is RAS-MAPK. In this pathway, upon HGF binding to c-MET, the adaptor protein GRB2 binds to phosphorylated tyrosine residues on c-MET via its SH2 domain and subsequently recruits SOS (Son of Sevenless) through its SH3 domains [18]. SOS is then able to activate RAS through GDP/GTP exchange, triggering the MAPK cascade, whose effects in the nucleus include gene expression regulation. MAPK signaling in the nucleus activates transcription factors such as Elk1, Etsl, and c-Myc, thereby influencing proliferation, differentiation, and survival [18]. Aberrant activation of the MAPK pathway has been implicated in promoting tumor cell invasion and metastasis, and its inhibitors can suppress tumor growth and progression [19]. The second pathway is the PI3K-AKT pathway. The second pathway is the PI3K-AKT pathway. Upon activation, c-MET serves as a docking site for the SH2 domain of p85, a regulatory subunit of PI3K. This leads to the activation of PI3K, which phosphorylates phosphatidylinositol 4,5-bisphosphate (PIP2) producing phosphatidylinositol (3,4,5)-trisphosphate (PIP3) [18,20]. There are two ways by which p85 can be recruited to c-MET: it can either bind directly to c-MET or indirectly through GAB1, both of which will signal to AKT [18]. Specifically, PIP3 activates AKT, which promotes cell survival and growth by inhibiting pro-apoptotic factors (e.g., BAD, FOXO) and activating downstream effectors involved in metabolism, protein synthesis, and cell cycle progression (e.g., mTOR, GSK3β, and MDM2) [18,21,22,23]. Finally, the third pathway is the JAK-STAT pathway, in which c-MET indirectly activates JAK or SRC kinases. This activation leads to the phosphorylation of STAT proteins. Phosphorylated STATs then dimerize and translocate into the nucleus, where they regulate the transcription of genes involved in cell differentiation, metabolism, survival, homeostasis, and immune response. This pathway plays a critical role in cell signaling, influencing various cellular processes that are essential for normal cellular function and tumorigenesis [24,25].

Endosome signaling and sorting play crucial roles in facilitating downstream signaling pathways following HGF binding to c-MET. Upon ligand binding, c-MET undergoes endocytosis and traffics to peripheral endosomes, where it continues to signal. During endosomal signaling, c-MET remains bound to HGF [26,27], which is essential for promoting cell migration. The specific downstream signaling cascade activated by c-MET is highly dependent on the type of endosome to which the receptor-ligand complex is localized. For example, activation of ERK1/2 requires trafficking of the c-MET-HGF complex to early endosomes and from there, ERK1/2 can translocate to other complexes to complete signaling, which has been associated with enhanced wound healing [26,28].

Adding to the complexity of this signaling network, vesicular trafficking also mediates a reciprocal interaction between matriptase and EGF signaling, which contributes to cancer progression. Matriptase, a membrane-anchored protease implicated in skin and breast cancer, undergoes EGF-induced endocytosis together with the EGF receptor. In endosomes, matriptase is activated by acidification and is then secreted extracellularly via exosomes, where it catalyzes cleavage of hepatocyte growth factor precursor (pro-HGF), initiating autocrine HGF/c-MET signaling. This matriptase-induced HGF/c-MET signaling constitutes a second wave of EGF signaling, promoting cancer cell scattering, migration, and invasion [29].

Furthermore, c-MET internalization and trafficking to a perinuclear compartment is required for efficient STAT3 activation and nuclear accumulation. While c-MET and STAT3 can interact at the plasma membrane, this interaction is weak due to rapid dephosphorylation by cytoplasmic phosphatases [28]. But, when c-MET is properly trafficked to the perinuclear compartment, significant STAT3 phosphorylation can occur, resulting in efficient nuclear accumulation and enhanced transcriptional activity.

Reactive oxygen species (ROS), particularly H_2_O_2_, can also induce retrograde trafficking of full-length c-MET from the plasma membrane to the nucleus in breast cancer cells. This transport is mediated by COPG1 and SEC61β and occurs independently of c-MET kinase activity. Nuclear c-MET has been shown to interact with DNA repair proteins, indicating a potential role in ROS-induced genomic maintenance and therapeutic resistance [30]. In addition, c-MET signaling is intricately linked to integrin pathways, which together regulate processes such as cell adhesion, migration, and survival [31]. Integrins, as key adhesion receptors mediating cell-extracellular matrix (ECM) interactions, can influence and be influenced by c-MET signaling through two principal modes: inside-out and outside-in signaling. Inside-out signaling involves intracellular signals activating integrins by inducing conformational changes in their extracellular domain [31]. Conversely, outside-in signaling has been observed through the phosphorylation of c-MET upon cell adhesion in the absence of HGF [31,32,33]. Notably, knockdown or blockade of specific integrins, such as α5β1, via siRNA significantly reduces c-MET phosphorylation, supporting the notion that integrins can function upstream of c-MET in certain contexts [34].

While the precise molecular mechanisms by which integrins regulate c-MET activation remain unclear, it is hypothesized that the cytoplasmic domains of integrins play a critical role. Moreover, integrins may also function downstream of c-MET as signaling adaptors, further emphasizing the bidirectional nature of c-MET–integrin crosstalk [35].

## 3. Genomic Alterations of MET Gene

### 3.1. MET Gene Amplification

Aberrant activation of c-MET signaling by *MET* gene amplification, mutation, or overexpression can lead to migration, invasion, proliferation, neo-angiogenesis within tumors, and metastasis of cancer cells. A common *MET* alteration is gene amplification. Genomic amplification of *MET* is frequently observed in human cancers, including non-small cell lung cancer (NSCLC), where it occurs in 25–75% of cases, depending on the study and criteria used to define overexpression [36,37]. In NSCLC, *MET* gene amplification is found in approximately 4% of patients, primarily in those with adenocarcinoma who have not previously been exposed to systemic therapies. Additionally, *MET* amplification is observed in up to 20% of patients with acquired resistance to tyrosine kinase inhibitors (TKIs), particularly following treatment with epidermal growth factor receptor (EGFR) inhibitors [36,38,39,40]. High *MET* gene copy number was significantly associated with poor survival, as patients with a mean of five or more *MET* gene copies had a significantly worse survival than individuals with lower copy numbers [41]. High *MET* copy number also correlates with response to c-MET inhibitors. In a study conducted by Kawakami et al., gastric cancer cell lines were examined for *MET* amplification. It was found that cell lines negative for *MET* amplification had *MET* copy numbers ranging between 1.3 and 3.3, whereas *MET* amplification positive cell lines had copy numbers ranging between 17.9 and 21.3 copies [42]. Further examination and analysis using the annexin-v binding assay and immunoblotting analysis found that cancer cells with *MET* amplification are more dependent on the HGF/c-MET signaling pathway for oncogenic growth [42]. As a result, these cancer cells respond well to c-MET inhibitors, highlighting c-MET as a target for drug development. It is also essential to highlight that while higher levels of *MET* amplification tend to correlate with a better response to c-MET inhibitors, overall responses in MET amplified tumor are low, suggesting that wt c-MET may be a poor oncogenic driver even when overexpressed.

### 3.2. MET Exon 14 Skipping Mutations

The first reports of *MET* exon 14 mutations came from primary lung cancer samples and cell lines [43,44]. More than 500 different mutations have since been reported to cause MET exon 14 skipping (*METΔEx14*), commonly due to point mutations or deletions that disrupt the splice donor or acceptor sites flanking exon 14. As mentioned earlier, exon 14 encodes the JM region, which includes the Y1003 residue crucial for negative regulation. Skipping of exon 14 removes Y1003, a key phosphorylation site required for CBL-mediated c-MET ubiquitination and subsequent degradation, resulting in aberrant c-MET accumulation and prolonged signaling, thereby driving unchecked cellular proliferation [45,46,47,48]. In other words, the *METΔEx14* protein lacks a key mechanism that turns off the protein once its function is fulfilled. Deletion of the JM domain also alters the conformation of the c-MET receptor, leading to constitutive activation of the kinase domain. As such, the skipping of exon 14 leads to sustained c-MET signaling and contributes to tumorigenesis [49,50]. Figure 3 illustrates the normal splicing of the *MET* gene, aberrant splicing resulting in exon 14 deletion, and the corresponding changes in *MET* transcripts and c-MET protein before and because of the mutation.

*METex14* mutations are detected most frequently in NSCLC [51,52,53]. A systematic literature review conducted by Mazieres reported a median *METex14* skipping frequency of 2.0% in unselected NSCLC patients, 2.4% to 2.6% in adenocarcinoma or non-squamous subgroups, 1.3% in tumors with squamous histology, and 12.0% to 31.8% in sarcomatoid carcinoma [52,53]. Typically, patients with tumors harboring *METex14* skipping mutations are age 70 or older and often have a previous history of or exposure to tobacco usage [54,55,56,57]. In a study by Kong-Beltran et al. [58] investigating the effects of *METΔEx14* on downstream signaling upon HGF stimulation, elevated phosphorylation levels of c-MET, MAPK, and AKT were observed in NSCLC cell lines. Cell lines with wild-type (wt) MET exhibited a decline in phosphorylation levels of c-MET and components of the MAPK pathway over time. In contrast, phosphorylation levels of *METΔEx14* remained stable for up to 3 h following HGF stimulation, resulting in an overactivation of c-MET-dependent pathways, whereas in the wt allele cell lines, they exhibited a steady loss of phosphorylation activity over this period. It was also found that the mutant form, over the wt form, was the predominant form of the protein expressed in tumor samples, despite the tumors being heterozygous for the exon 14 deletion [59]. Additional studies have been conducted to confirm the overexpression of the deleted form. A study that analyzed c-MET expression ratios in both *METΔEx14*-negative and *METΔEx14*-positive samples found that the expression of the wt allele was significantly lower in *METΔEx14*-positive samples compared to control samples that still had exon 14 intact [59].

Notably, tumors with *MET* exon 14 deletion mutations do not harbor mutations in other common proto-oncogenes, as *METΔEx14* were generally wt for *KRAS, BRAF*, and *EGFR*. Specifically, in The Cancer Genome Atlas (TCGA) database, approximately 10% of lung adenocarcinoma patients with wt *EGFR, RAS, RAF* may carry a *MET* allele that results in exon 14 skipping [60], suggesting that this mutation is mutually exclusive with mutations in other known proto-oncogenic genes. Additionally, *METex14* deletion was not found in normal human lung samples, indicative of a dominant role of the mutant c-MET in driving tumor development [58]. Importantly, both *METΔEx14* mutations and MET amplification correlate with poor prognosis [44]. Also, both forms of *MET* alteration can be found together, with the rate of co-occurrence ranges from 0% to 40.5%, further supporting c-MET alteration as a therapeutic target.

### 3.3. MET Missense Mutations

*MET* missense mutations have been identified in several functional domains, including the kinase, JM, and SEMA domains [56]. The first activating mutations identified were germline mutations in the kinase domain from hereditary papillary renal carcinomas (HPRC) [61]. These mutations (M1149T, V1206L, V1238I, D1246N, and Y1248C) flank the critical tyrosine residues Y1234 and Y1235 within the kinase domain [58,61]. Missense mutations in this domain were also identified in childhood hepatocellular carcinoma (HCC) [62], advanced head and neck cancers [63], sporadic papillary renal carcinoma (SPRC) (ref.), and familial colorectal cancer [56,64]. Notably, there are reports of clinical responses to crizotinib in patients with MET H1112Y (also referred to as MET H1094Y) mutant papillary renal cancer, demonstrating the potential therapeutic implications of targeting these mutations [65]. In tumors with *MET* mutations affecting the ATP-binding site, sustained expression of the mutated gene variant was observed. In cell lines, mutations such as N1118Y and those affecting V1110, and H1112 result in amino acid class changes in the ATP-binding region on the receptor, assisting in its activation [66]. Like the loss of CBL binding caused by exon 14 skipping, the mutations in the ATP-binding site are highly associated with constitutive phosphorylation of the receptor and sustained downstream signaling, contributing to tumor progression. Moreover, somatic mutations such as Y1230C and Y1235D are linked to enhanced kinase activity, likely by mimicking that of a phosphorylated tyrosine [67].

Missense mutations in the JM domain and the SEMA domain have also been identified in gastric cancer, breast cancer, small cell lung cancer (SCLC) and NSCLC [43,44,55]. Such mutations have been shown to reduce c-MET receptor ubiquitination and degradation, thereby extending its signaling duration, while SEMA domain mutations are likely to impact the structure of the ligand-binding domain [68].

### 3.4. MET Gene Fusions

*MET* fusions are rare oncogenic driver events in cancers. The first *MET* gene fusion, a *TPR-MET* fusion, was first identified in a human osteosarcoma cancer cell line treated with a chemical mutagen [69]. TPR, or translocated promoter region, is a protein encoded by a gene on chromosome 1, while *MET* is on chromosome 7. Transcript of the fusion gene reveals that at least two exons of the fusion gene are from *TPR*, and the *MET* portion possibly uses the *TPR* promoter. The fusion likely uses the TPR dimerization motif, allowing constitutive signaling and ligand-independent activation of the mutated c-MET kinase. Since this discovery, several additional *MET* gene fusions have been observed, particularly in pediatric glioblastomas. *MET* fusions can be separated into at least 2 classes. There are fusions, such as where the N-terminal of *MET*, most often including exon 14, has been replaced with a partner that contains a dimerization motif. Such fusions, such as *KIF5B-MET* are often primary drivers, are oncogenic and sensitive to MET inhibitors in vitro [70]. Another common class of fusions is where early exons of a partner gene that is mostly UTR is rearranged to intron 1 of MET. As the coding region of MET starts in exon 2, this would lead to “promoter hijacking”, where the full coding region of MET is placed under control of a different promoter region. Examples of this include the PTPRZ1-MET fusion seen in a subset of pediatric glioblastomas [71]. These fusions may be less oncogenic, theoretically functioning just to overexpress wt MET. However, it is interesting that glioblastomas expressing PTPRZ1-MET often also express *METΔEx14* splice variant, suggesting that MET activation in these tumors may be complex. Most recent studies showed that *MET* fusions occur in multiple solid tumors, but are overall rare, for example, being seen in 0.2% to 0.3% of lung cancers [72,73]. The currently known fusion genes observed in patient tumor samples from the University of Texas Austin database are summarized in Table 1. Notably, while the MET fusion genes listed in the table were identified in tumors, their functionality and roles in tumorigenesis remain largely unverified.

Figure 4 illustrates the distribution of cancer types observed in patients with *MET* gene fusions. It shows a range of cancers where *MET* gene fusions were present, detailing the prevalence of these fusions across different tumor types. The distribution pattern helps in understanding the role of *MET* gene fusions in different cancer types and their potential impact on tumor progression and treatment strategies.

While SVA insertions have been implicated in transcript fusions and alternative splicing in various cancers, no direct association with the *MET* gene has been reported to date. However, given MET’s complex transcriptional regulation and frequent involvement in oncogenic rearrangements, further studies may uncover potential interactions [78].

## 4. Variation in Tumor Profiles

While *MET* exon 14-skipping mutations are found in many types of cancer, such as gliomas, gastrointestinal cancers, and sarcomas, they are predominantly found in lung cancers and most commonly in NSCLCs. About 32% of pulmonary sarcomatoid lung carcinomas harbor exon 14 alterations, and approximately 3–4% of lung adenocarcinomas contain an alteration that leads to skipping of exon 14 [43,51,79]. Within various subtypes of lung cancer, those with an adenocarcinoma component were more likely to harbor these mutations compared to those without [44]. Additionally, *MET* JM domain mutations were found in lung but not in colon cancers, suggestive of a tumor site specificity [58]. As for c-MET’s oncogenic ability, it was found to be similar to or greater than that of other known NSCLC oncogenic drivers, such as ROS1, RET, BRAF, and ALK [80,81,82,83,84].

Outside the scope of lung cancer, it was also found that different *MET* exon 14 mutations predispose to diverse tumor types [85]. On top of those mutations, mice harboring mutations such as D1226N, Y1228C, M1248T and M1248T/L1193 double mutation more frequently developed sarcomas and lymphomas, whereas mice carrying the M1248T single mutation often developed carcinomas and lymphomas [85]. Mutant c-MET was also found to be associated with the aggressive phenotype of osteosarcomas. In a study conducted by Zeng et al., in 12 osteosarcoma cell lines, the c-MET/HGF receptor exhibited full functionality but experienced constitutive phosphorylation. Activation of c-MET/HGF in osteosarcomas may contribute to downstream signaling pathways involved with motility, mitogenesis, and morphogenesis in an aggressive manner [86].

In preclinical and clinical studies, the HGF/c-MET signaling pathway has been found to play a significant role in breast cancer tumorigenesis and disease progression, contributing to the invasive phenotype seen in many tumors. Across all types of breast cancers, at least 20–30% of cases show c-MET overexpression [87], and HGF and c-MET are often co-expressed in invasive breast cancers [88,89,90,91]. While c-MET protein overexpression is frequently observed in various cancers and linked to poor prognosis, its functional relevance in the absence of a genomic alteration (like gene amplification or mutation) remains a subject of ongoing research and is not fully understood, through either paracrine or autocrine signaling, the pathway is important for breast cancer progression. From a prognosis standpoint, higher levels of c-MET and phospho-c-MET in breast cancers correlate with worse prognosis [92,93,94]. In a study conducted by de Melo Gagliato et al. [93], it was found that all tumors harboring *MET* gene amplification and 87.5% of tumors with *MET* mutation were high-grade tumors, while only about 56% tumors without *MET* amplification or mutation in the same cohort were of high grade, demonstrating the markedly negative impact of *MET* alterations on patient outcome. *MET* amplification has been associated with the metastatic spread of EGFR-mutated NSCLC [95]. Similarly, in breast cancers, it was also found that *MET* mutation or amplification promotes tumor metastasis into more diverse organs; the median number of organs involved for the group of tumors with non-amplified *MET* was 2, compared to 7 for the MET amplified group. While the finding may not be considered conclusive due to the small sample size examined, this study suggests a potentially key role for c-MET in breast cancer metastasis. In another study, Garcia et al. examined over 900 breast cancer specimens and found an association between amplified *MET* and features of basal-like cancers [96]. Through further examination, they argued that *MET* amplification could be a key component and preferentially expressed in basal-like breast cancer (BLBC), an aggressive subtype of breast cancer [97,98]. Moreover, a separate study revealed that among all breast cancers, the basal subtype was correlated with the highest levels of c-MET in tumor samples [99]. Additional evidence from Beviglia et al. indicates that BLBCs are not only characterized by *MET* over-expression and amplification, but also by elevated levels of c-MET phosphorylation [100,101,102]. Specifically, the authors examined c-MET receptor expression in both tumor samples and cultured cancer cells, and they found that poorly differentiated cell lines frequently expressed high levels of the receptor and that high c-MET levels were associated with increased motility and invasiveness [100].

## 5. Drug Treatment for MET Genomic Alterations

Due to MET’s complex regulations, there are many drug and drug candidates in various phases of the development. Preclinical studies have shown that c-MET-specific blockade was only effective in tumors harboring *MET* genomic alterations, whereas targeting the c-MET pathway in the setting of wt *MET* had little effect on cancer growth [103]. Thus, it is crucial to understand *MET* gene status before beginning treatment. Most c-MET targeting agents have been tyrosine kinase inhibitors, or TKIs, and they can be classified into type 1, type 2, and type 3. Both type 1 and 2 agents are ATP-competitive inhibitors, while type 3 agents are allosteric inhibitors [104]. Type 1 inhibitors interact with the ATP-binding pocket in the active state and are the most common TKIs used for tumors with *MET* exon 14 mutations. Type 2 inhibitors bind to the inactive form of the kinase, in which the ATP-binding pocket is slightly more open than that in the active form, and extend into the hydrophobic back pocket that is inaccessible when the kinase is in its active form [104]. Type 1 inhibitors are divided into two subgroups, type 1a and type 1b: type 1a inhibitors interact with both Y1230 and G1163 residues, while type 1b interacts with Y1230 but not G1163. A current type 1a inhibitor is Crizotinib, whereas type 1b inhibitors include Capmatinib, Tepotinib, and Bozitinib [104]. Although various drugs for cancers harboring exon 14 deletions are currently tested only in NSCLC, increasing amount of evidence from clinical studies have shown that patients with other tumors harboring a *MET* mutation would also benefit from these targeted therapies [104]. Specifically, in vitro studies have shown that c-MET-targeting agents can inhibit the oncogenesis caused by exon 14 loss, providing a solid framework for future therapeutic development [47,58]. However, the current results and efficacy data must be viewed with caution, as varying definitions of gene amplification across different clinical trials can complicate the interpretation of results from available c-MET inhibitors [105]. Additionally, none of the selective c-MET inhibitors currently on the market is significantly more effective than the others [105]. Despite these challenges, current drug treatments show promising results and highlight an area of ongoing development for future therapies. As shown in Figure 5, the schematic representation illustrates the mechanism of action of c-MET inhibitors.

### 5.1. Crizotinib: A Multitarget Inhibitor

Crizotinib, or XALKORI, is a type 1a inhibitor. Originally designed to inhibit ALK phosphorylation by interacting with the Y1230 residue and causing G1/S-phase cell cycle arrest and apoptosis [106], Crizotinib also has potent activity against c-MET with exon 14 alterations. Crizotinib has had recent success in clinical responses in patients with lung adenocarcinoma [107,108,109]. Crizotinib is also approved for cancers with ALK or ROS1 alterations [106]. Currently, it is a treatment option for NSCLC patients who have *METΔEx14* mutations. In a study conducted from 2014 to 2018, 69 patients with NSCLCs containing *METΔEx14* alterations were treated with Crizotinib; 62% of patients had received at least one previous line of treatment. In total, 65 of the 69 patients had evaluable tumors, and the objective response rate (ORR) was 32% (95% confidence interval (CI), 21–45). Among the treated patients, 5% had a complete response, where treatment removed the entirety of the tumor, 28% had a partial response, and 45% had stable disease as their best overall response. In the PROFILE 1001 study examining the activity of Crizotinib, patients were also split into categories based on *MET* amplification copy number. The high-*MET*-amplification group, defined as more than 4 copies of *MET*, saw the highest ORR of 38.1% and a median progression-free survival of 6.7 months, compared to the low-MET amplification group (less than 2.2 copies) that had a median progression-free survival of 1.8 months. It is important to consider this unique response to Crizotinib, as this drug is more effective for patients with high *MET* copy numbers. It was also noted that a decrease in lesion size was observed in most patients. Additionally, the study showed that responses were rapid and durable, with 57% of responders having a response lasting at least six months after drug treatment [110,111]. Compared with second-line chemotherapy and other traditional treatments, the observed outcomes showed Crizotinib had superior responses [110]. However, compared with outcomes in NSCLC harboring other targetable oncogenes, the ORRs and duration of response for Crizotinib were lower [110]. Of note crizotinib has poor CNS penetration, and development of CNS mets on treatment can occur. While most studies on c-MET-targeted therapies have focused on NSCLC, emerging evidence suggests that patients with other tumors harboring *MET* mutations may also benefit from these treatments. A notable example is a study by Ganesan et al. [65], which explored the response to Crizotinib in a patient with *MET*-mutated papillary renal cell carcinoma (PRCC) who had progressed on Tivantinib. The case specifically demonstrated that Crizotinib was able to induce a significant response in the patient despite previous progression on Tivantinib and underscoring the therapeutic value of c-MET inhibition in diverse tumor types beyond NSCLC.

### 5.2. Capmatinib and Tepotinib: Selective Inhibitors

Type 1b inhibitors include Capmatinib, Tepotinib, and Bozitinib. Capmatinib or Tepotinib are currently the preferred treatment for metastatic NSCLC with *METΔEx14* mutations. Capmatinib and Tepotinib are designed as a selective small-molecule inhibitors that indirectly prevent the activation of downstream signaling by blocking c-MET phosphorylation at the start, by binding to the ATP-binding pocket of the kinase in its active conformation [112]. Capmatinib is highly specific against c-MET, with more than 10,000-fold selectivity over other kinases [103]. A phase 2 study investigating the efficacy of Capmatinib in NSCLC patients found that patients who had not received previous lines of treatment responded better than those who had, with ORRs of 40% (95% CI, 16 to 68) and 29% (95% CI, 19 to 41), respectively [113]. Additionally, Capmatinib showed high intracranial activity for brain metastasis and favorable safety assessment, as most adverse events were only grades one and two [103]. In an evaluation performed among 13 patients in the GEOMETRY mono-1 trial, about 92% of patients had intracranial disease control and 54% had an intracranial response, with 31% of patients showing a complete response to Capmatinib [113,114]. Additionally, there was a difference in efficacy and response between tumors with high vs. low or no *MET* amplification, as tumors with less than 10 copies of *MET* responded poorly to Capmatinib treatment, and those with ≥10 copies having an ORR of 29%. A clinical trial performed by Paik et al. treated 99 patients with Tepotinib; the ORR was 46% (95% CI, 36 to 57) and all responses were partial, with no patient showing a complete response. Additionally, no intracranial response rates have been reported [115]. Response rates were similar regardless of the number of previous lines of therapy, distinguishing Tepotinib from other approved treatments. Patient quality of life was maintained during the Tepotinib trial, making it a strong candidate for safe treatment [116]. However, close monitoring of treatment-related adverse events (TRAE) to c-MET inhibitors is necessary, as older participants (above 75 years old) had more frequent incidents of higher-grade (grade 3 or higher) TRAE than compared with younger patients (33.9% versus 18.5%), with peripheral edema being the most common [105]. Both Capmatinib and Tepotinib received FDA approval in 2020 and 2021, respectively, under accelerated approval, and are currently considered first-line therapies for NSCLC patients with *METΔEx14* mutations [117]. On the other hand, Bozitinib, a highly selective and specific c-MET inhibitor that has shown promising efficacy in mouse models [104], is still in phase two. Mouse studies have revealed its potentially higher permeability and lower efflux rate compared with the other type 1b drug treatments on the market, fostering a hopeful outlook in future treatment options for *METΔEx14* patients [104].

### 5.3. Type 2 and 3 Inhibitors

Type 2 inhibitors interact with the kinase to trap it in its inactive conformation by exploiting a hydrophobic back pocket adjacent to the ATP-binding site [106]. As a result, there is potentially greater kinase specificity and slower dissociation from the drug binding site for type 2 inhibitors compared to type 1 [118]. Examples of type 2 inhibitors include Cabozantinib and Merestinib. Cabozantinib is a TKI with activity against a wide range of targets, including MET, VEGFR2, and FLT3 [119]. It was approved in 2012 by the FDA for the treatment of medullary thyroid cancer and renal cell cancer. In mouse models using nude mice, xenografts showed reduced cell proliferation and increased apoptosis when treated with Cabozantinib. In 2012, 300 patients were enrolled in the EXAM double-blind placebo-controlled clinical trial. Patients of varying previous treatment levels were considered in this trial. The primary endpoint was progression-free survival (PFS) and was met with a median of 7.2 months PFS compared to 4.0 months PFS in the placebo group [120]. Regarding one-year PFS, 47.3% of patients reached this benchmark compared to 7.2% in the placebo group. In another study (METEOR trial), Cabozantinib was studied in the context of patients who had at least one previous line of VEGFR-directed therapy. The trial consisted of 658 patients and used Everolimus, an mTOR (kinase) inhibitor, as its control. When treated with Cabozantinib, there was an ORR of 21% and overall survival improving to 21.4 months, compared to baseline 5% ORR and 16.5 months of overall improved survival with Everolimus treatment [121,122]. The side effects of Cabozantinib are like other TKIs, with diarrhea, fatigue, and hypertension reported as the most common side effects [123]. The results of these two studies indicate Cabozantinib to be a promising TKI for MET-related cancer treatment, with substantial improvement in PFS and overall survival in patients with varying amounts of previous treatment. Currently, Cabozantinib is being studied for combination therapy with immune checkpoint inhibitors, undergoing phase 1 clinical trials [124]. Merestinib is also a type 2 inhibitor of MET, with antitumor proliferative and antiangiogenic activity in MET-amplified tumor models and also designed to be active against other receptor tyrosine kinases, including AXL, ROS1, and FLT3. [125]. In a phase 1 dose-escalation clinical trial, 186 patients with advanced cancers were enrolled, receiving Merestinib at doses ranging from 20 mg to 180 mg daily. The trial reported that 32% of patients achieved stable disease as their best overall response, while 52% experienced progressive disease. The study concluded that Merestinib has a tolerable safety profile and potential anticancer activity, warranting further clinical investigation [125]. With respect to treatment-emergent adverse events (TEAEs), 96% of patients experienced TEAEs of varying grades and 40% experienced higher than grade 3 TEAEs [125]. It is not yet approved by the FDA and is still in clinical trials. Subsequent phase II trials, did not show significant improvement in progression-free survival or terminated early due to adverse events and other factors, leading to a decrease in the drug’s likelihood of approval [126]. Type 3 c-MET inhibitors are still in very early stages of development and not as well understood.

### 5.4. Antibody-Based Treatments

Aside from traditional small molecule drugs, antibodies and other immunotherapy options have also been explored with NSCLC patients containing *METΔEx14* mutations. Instead of directly targeting the ATP-binding site in the kinase domain of c-MET, antibodies block the interaction between HGF and c-MET [104]. A preclinical study conducted by Glodde et al. suggests a potential role for the HGF/c-MET pathway in neutrophil recruitment, outlining a new path for drug development [127]. Currently, there are two main antibody treatments in phase two and first-in-human stages of development: Sym-015 and Telisotuzumab vedotin, respectively. Sym-015 is a mixture of two humanized IgG1s, Hu9006, and Hu9338, which targets the SEMA domain, preventing the binding of HGF to c-MET [104]. By blocking the HGF-c-MET interaction, it induces receptor internalization and degradation, while also stimulating complement-dependent cytotoxicity (CDC) and antibody-dependent cell-mediated cytotoxicity (ADCC) in vitro and in vivo. It has been shown to effectively inhibit the growth of *MET*-amplified tumors [128,129]. Telisotuzumab vedotin is an antibody-drug conjugate (ADC) consisting of a monoclonal antibody against c-MET conjugated to monomethyl auristatin E (MMAE), a potent inhibitor of tubulin polymerization. When enzymatically cleaved, Telisotuzumab vedotin releases MMAE into the cytosol, causing cell cycle arrest at the G2/M phase and ultimately contributing to tumor cell apoptosis [104]. In a first-in-human trial, 16 patients were eligible for treatment with Telisotuzumab vedotin. Of these 16, 3 (18.8%; 95% CI: 4.1–45.7) achieved a partial response. Building on these early findings, the phase II LUMINOSITY trial (NCT03539536) assessed Teliso-V in 172 patients with nonsquamous EGFR-wildtype NSCLC and c-MET protein overexpression. The patients were stratified into high (≥50%) and intermediate (25–49%) c-MET expression groups. Teliso-V, administered at 1.9 mg/kg every two weeks, demonstrated an overall response rate (ORR) of 28.6%, with notably higher efficacy in the high-expression group (34.6% vs. 22.9%). The median duration of response was 8.3 months, and overall survival reached 14.5 months. These findings underscore Teliso-V as a promising treatment strategy, particularly for tumors with high c-MET expression, offering durable clinical benefit with manageable toxicity—most notably grade ≥3 peripheral sensory neuropathy in 7% of patients [130]. Responses to immunotherapy have not been consistent and remain an area for more extensive study. As of now, specifically for *MET* amplification and exon 14 deletion mutations, primarily in vitro studies have been conducted using immunotherapy techniques in gastric cancer cell lines, and further research is needed to achieve conclusive results in vivo and in clinical settings [104].

Another antibody-based treatment that is currently an ongoing phase 1 dose escalation and dose expansion study is Amivantamab [131]. Amivantamab is a human, bispecific antibody that targets EFGFR and c-MET. It is currently approved for the treatment of NSCLC patients with an EGFR exon 20 insertion mutation. Given that Amivantamab is a bispecific antibody, it can simultaneously bind to two different antigens or epitopes, making it a potential *METΔEx14* drug candidate. It is therapeutic effects on cancers with *METΔEx14* deletions are currently being explored in the MET-2 cohort of the CHYRSALIS study. In this cohort of the study, Krebs et al. examine its effects on *METex14* NSCLC patients that have undergone a range of previous lines of treatment, including Tepotinib, Capmatinib, and no previous lines of therapy. So far, the overall response of these patients to Amivantamab is 33% (50% in patients with no previous lines of therapy, 46% in patients with no prior c-MET inhibitor, and 21% in patients with prior c-MET inhibitor therapy). When assessing median duration of response, about 67% of patients had a duration of response to Amivantamab of at least 6 months or more [131]. The safety profile of Amivantamab in *MET*-exon 14 deletion patients was like previous reported experiences in *EGFR*-exon 20 insertion patients. At this point of the phase 1 study, Amivantamab has demonstrated anti-tumor activity in *METex14* NSCLC patients, although the study is still ongoing, and new cases are still being assessed. Although it is still ongoing, it has shown promise in both treatment-naive patients and those who have previously undergone therapy, positioning Amivantamab as a potentially effective antibody-dependent treatment for cancers with MET alterations [131]. A summary of c-MET inhibitors is shown in Table 2.

### 5.5. Mechanisms of Resistance

Like other therapies, cancer cells can develop resistance to c-MET-targeting therapies through a variety of mechanisms, including both on- and off-target mechanisms. In exon 14-mutated NSCLC, secondary mutations have been identified in the kinase domain that are associated with resistance to c-MET inhibitors. For example, secondary mutations in the A-loop residues D1228 and Y1230 have been found to increase resistance to type 1 c-MET inhibitors by altering ATP binding in the kinase domain [133,134,135]. Although these two missense mutations impact responsiveness to type 1 MET inhibitors, they do not appear to affect type 2 inhibitors. Switching to a type 2 inhibitor after acquired resistance to a type 1 inhibitor may lead to clinical response [136,137]. However, treatment using type 2 inhibitors on cancer cells that have acquired type 1 inhibitor resistance may also not always be effective, as seen in Recondo et al. [138]. For example, certain mutations in the c-MET receptor can cause conformational changes that prevent both type 1 and type 2 inhibitors from binding effectively, limiting their therapeutic benefit. Additionally, activation of alternative signaling pathways (such as EGFR or HER2) can bypass c-MET inhibition altogether, rendering type 2 inhibitors ineffective. Off-target mechanisms may involve mutations that enable cancer cells to bypass c-MET inhibition by activating alternative pathways, sustaining proliferation and metastasis signaling despite the inhibition. This includes KRAS mutations and amplifications of other signaling kinases including *EGFR*, *HER3*, and *BRAF*. Given the challenges of acquired resistance, it is crucial to comprehensively profile the tumor both before and during treatment, as conditions may change rapidly. Approximately 3–4% of NSCLC patients harbor *METex14* mutations and given the need to assess multiple biomarkers in NSCLC, single-gene testing is insufficient for comprehensive profiling [139,140,141,142,143,144]. Especially in NSCLC patients, the limited amount of tumor tissue in biopsies often restricts the scope of testing, prompting researchers to explore alternative diagnostic approaches, such as liquid biopsy [140,142,143,144]. Liquid biopsy is a non-invasive method that enables dynamic monitoring of circulating tumor DNA (ctDNA), providing insights into tumor heterogeneity and genomic alterations across different sites, including the primary tumor and off-target metastatic lesions. It is particularly valuable when tumor tissue is scarce. Still, tissue biopsy has the advantage of being able to provide histopathological context [80]. Together, these approaches facilitate the monitoring of resistance mechanisms as they develop, offering researchers deeper insights into tumor progression and guiding patient treatment [103].

## 6. Conclusions

Alterations in *MET* have been identified as primary oncogenic drivers in cancer, particularly through mechanisms such as exon 14 skipping, missense mutations, or gene fusions. Gene amplification, especially focal, high-level amplification, may also serve as a driver in some settings. These alterations have been extensively profiled, with most studies focusing on their prevalence and impact in NSCLC. While current targeted therapies, such as TKIs, demonstrate promising responses, ongoing research into immunotherapies and antibody-drug conjugates are expanding the potential for innovative drug development. These emerging approaches hold the promise of addressing resistance to TKIs and improving outcomes in MET-driven cancers.

## Figures and Tables

**Figure 1 cancers-17-01493-f001:**
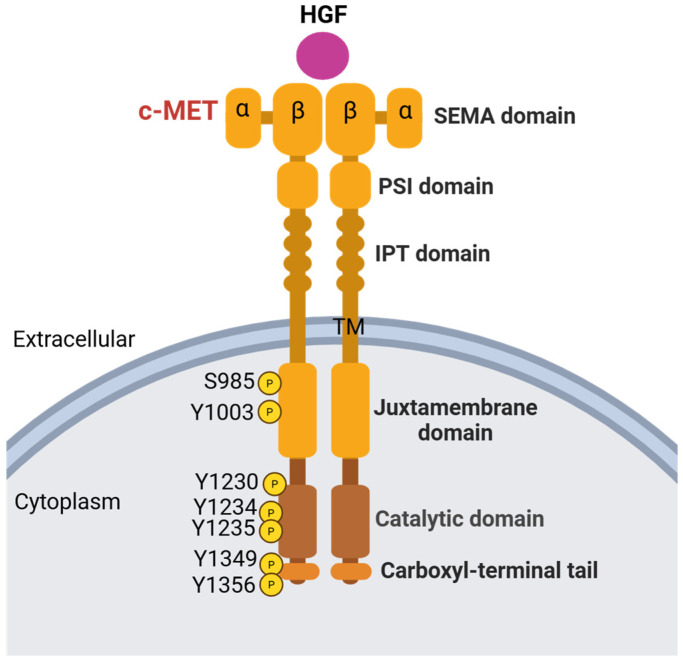
Structure and functional domains of the c-MET receptor kinase. c-MET consists of extracellular, transmembrane, and intracellular regions. The extracellular region includes a semaphorin (SEMA) domain, involved in ligand binding, and a PSI (plexins, semaphorins, and integrins) domain. Four immunoglobulin-like (IPT) repeats contribute to receptor stability and ligand interaction. The transmembrane (TM) region anchors the receptor in the cell membrane. The cytoplasm region contains the juxtamembrane (JM) domain, the catalytic kinase domain, and the carboxyl-terminal tail. Key phosphorylation sites involved in MET signaling: S985, Y1003 in the JM domain, Y1230, Y1234 and Y1235 in the catalytic domain, and Y1349 and Y1356 in the carboxyl-terminal tail are indicated. (Created in https://BioRender.com, accessed on 10 April 2025).

**Figure 2 cancers-17-01493-f002:**
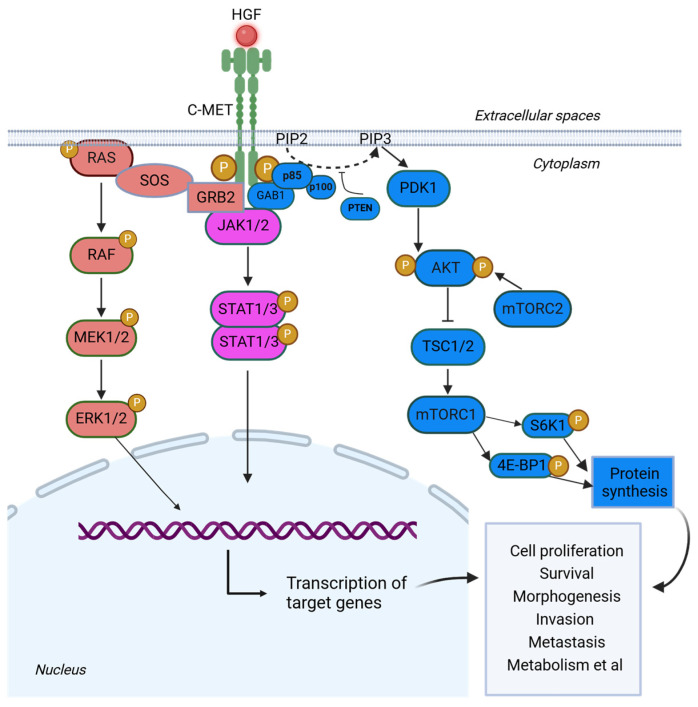
Signaling pathways activated by c-MET. Upon ligand binding, the c-MET receptor undergoes dimerization and autophosphorylation, which leads to the recruitment of adaptor proteins such as GAB1, GRB2, and other SH2-domain-containing proteins. This activation initiates several downstream signaling cascades, including the RAS-RAF-MEK-ERK pathway, which promotes cell growth and proliferation; the PI3K-AKT-mTOR pathway, which regulates metabolism, supports cell survival, and inhibits apoptosis; and the JAK-STAT3 pathway, which drives gene expression associated with cell proliferation, survival, morphogenesis, invasion, metastasis, and metabolism. These coordinated signaling events collectively contribute to oncogenesis and tumor progression by promoting pro-oncogenic gene transcription, cell survival, and metabolic adaptation. (Created in https://BioRender.com, accessed on 15 January 2025).

**Figure 3 cancers-17-01493-f003:**
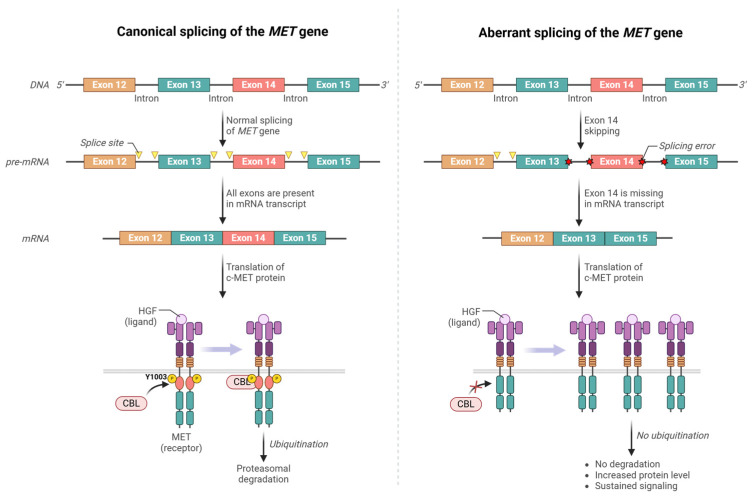
Impact of *MET* exon 14 skipping on receptor function. The left panel illustrates the normal splicing of *MET* pre-mRNA, which includes exon 14, resulting in the production of a full-length c-MET protein. This protein undergoes ubiquitination mediated by CBL, leading to proteasomal degradation and termination of c-MET signaling. The right panel shows the consequences of exon 14 skipping, which excludes exon 14 from the mature mRNA. This alteration produces a c-MET protein lacking the CBL binding site, preventing ubiquitination and degradation. Consequently, the c-MET receptor remains on the cell surface, leading to prolonged signal transduction and oncogenic activation. (Created in https://BioRender.com, accessed on 5 Feburary 2025).

**Figure 4 cancers-17-01493-f004:**
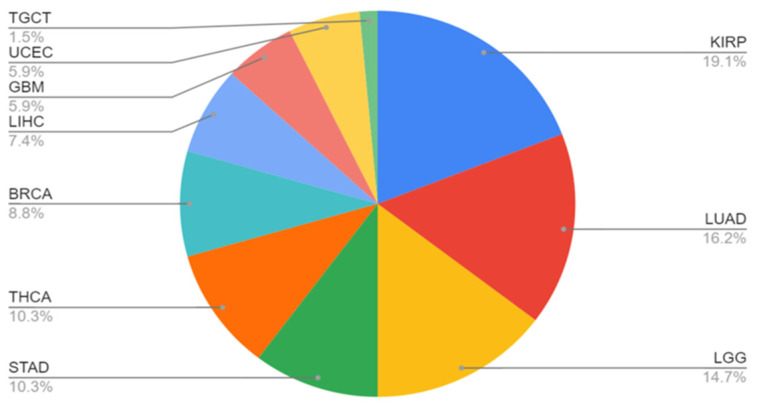
Distribution of MET gene fusions across different cancer types based on the fusion genes listed in Table 1. The pie chart displays the proportion of various cancer types harboring *MET* fusion genes. Each segment represents a different cancer type, with the size of each segment corresponding to the prevalence of *MET* fusions in that type. This distribution highlights the most common cancers associated with *MET* gene fusions. The abbreviations for tumor types are referenced from TCGA Study Abbreviations (https://gdc.cancer.gov/resources-tcga-users/tcga-code-tables/tcga-study-abbreviations, accessed on 5 September 2024). TGCT, Testicular Germ Cell Tumors; UCEC, Uterine Corpus Endometrial Carcinoma; GBM, Glioblastoma multiforme; LIHC, Liver hepatocellular carcinoma; BRCA, Breast invasive carcinoma; THCA, Thyroid carcinoma; STAD, Stomach adenocarcinoma; KIRP, Kidney renal papillary cell carcinoma; LUAD, Lung adenocarcinoma; LGG, Brain Lower Grade Glioma.

**Figure 5 cancers-17-01493-f005:**
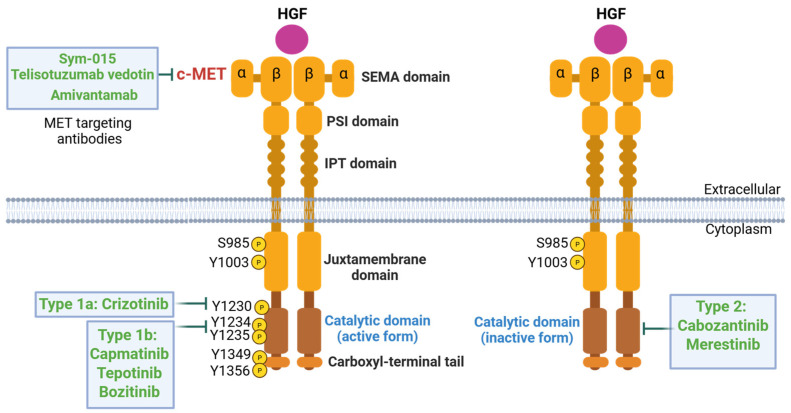
Mechanisms of action of MET-targeting therapies and classification of MET tyrosine kinase inhibitors (TKIs). The diagram illustrates the structure and inhibition of the MET receptor tyrosine kinase in its active and inactive conformations. On the left, MET is depicted in its active conformation, with phosphorylation of tyrosine residues in the intracellular kinase domain. Type I MET inhibitors (which bind the active conformation) are subdivided into Type Ia (e.g., Crizotinib) and Type Ib (e.g., Capmatinib, Tepotinib, Bozitinib), and they competitively block ATP binding in the kinase domain. On the right, MET is shown in its inactive conformation, targeted by Type II inhibitors (e.g., Cabozantinib, Merestinib), which bind an adjacent allosteric pocket only accessible when the activation loop is in an inactive state. Additionally, antibody-based therapies, such as Sym-015 and Telisotuzumab vedotin, target the extracellular SEMA domain of MET, blocking ligand (HGF) binding and inducing receptor internalization, degradation, or cytotoxic effects. Together, these therapeutic strategies highlight the diversity of approaches to inhibit aberrant MET signaling in cancer. (Created in https://BioRender.com, accessed on 8 April 2025).

**Table 1 cancers-17-01493-t001:** Characteristics of *MET* Fusion Genes and Corresponding Exon Breakpoints Across Cases.

Fusion Gene	Exons of Partner Gene	Exons of MET	Number of Cases
BAIAP2L1-MET *	9–14	15–21	3
BAZ1B-MET *	4–20	20–21	1
C8orf34-MET *	7–10	12–21	4
MET-C8orf34 *	3–14	14–21	4
CAPZA2-MET *	7–10	12–21	7
MET-CAPZA2 *	2–10	10–21	2
KIF5B-MET *	24–26	14–21	2
MET-CAV1 *	3	1–21	7
MET-CFTR *	23	1–21	4
MET-CNTNQAP *	3–24	2–21	1
MET-DYNC1I1 *	2–17	3–21	1
MET-TES *	2–7	2–21	1
TES-MET *	1–7	2–21	2
MET-TFG *	6–8	14–21	3
TFG-MET*	5–8	15–21	3
MET-WNT2 *	4–5	16–21	4
OXR1-MET *	11–16	13–21	4
PTPRZ1-MET * [74]	1–30	2–21	9
ST7-MET *	1–7	2–21	5
TPR-MET * [75]	1–30	15–21	16
TRK-MET * [76]	1–7	15–21	3
CLIP2-MET * [77]	1–11	15–21	2

Fusion genes data source: * https://ccsm.uth.edu/FusionGDB/, accessed on 8 April 2025.

**Table 2 cancers-17-01493-t002:** Summary of c-MET inhibitors.

Drug Name	Type Inhibitor	General Mechanism/Notes
Crizotinib	Type 1a	Inhibits ALK phosphorylation by interacting with the Y1230 residue, leading to G1/S-phase cell cycle arrest and apoptosis. More effective in patients with high MET copy numbers [104,106].
Capmatinib	Type 1b	A selective small-molecule inhibitor that blocks c-MET phosphorylation by binding to the ATP-binding pocket of the active kinase, thereby preventing downstream signaling [112].
Tepotinib	Type 1b	Similar to capmatinib; selectively inhibits c-MET phosphorylation by binding to its ATP-binding pocket in the active conformation [112].
Bozitinib	Type 1b	A highly selective and specific c-MET inhibitor currently in Phase 2 clinical trials [132].
Cabozantinib	Type 2	Multi-kinase; binds MET in inactive conformation, approved by the FDA in 2012; targets c-MET along with other receptor tyrosine kinases [119].
Merestinib	Type 2	Type 2 inhibitor with antitumor and antiproliferative effects against MET, also designed to be active against other receptor tyrosine kinases [125]
Sym-015	Antibody-based (IgG1 mixture)	A combination of two humanized IgG1 antibodies targeting the SEMA domain of c-MET to prevent HGF binding [127].
Telisotuzumab vedotin	Antibody-drug conjugate	Delivers monomethyl auristatin E (MMAE) to the cytosol, causing G2/M phase cell cycle arrest [127,130].
Amivantamab	Bispecific antibody	A human bispecific antibody targeting both EGFR and c-MET; induces receptor degradation and immune cell-mediated cytotoxicity. Under investigation for MET exon 14 alterations [131]

## Data Availability

The data used in this study for Fusion genes can be found at https://ccsm.uth.edu/FusionGDB/, accessed on 1 September 2024.

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
