# Peer review of "Targeting c-MET Alterations in Cancer: A Review of Genetic Drivers and Therapeutic Implications"

_cancers, 2025, doi:10.3390/cancers17091493_

Round 1

Reviewer 1 Report

Comments and Suggestions for Authors

The manuscript by Ji et al. highlights key alterations of the C-MET receptor across various cancer types and discusses its clinical significance as a therapeutic target. The C-MET receptor plays a crucial role in mediating several signaling pathways that regulate fundamental cellular functions such as survival, proliferation, and migration. Its oncogenic and tumorigenic signaling mechanisms, which significantly contribute to cancer development and progression, are well documented. The review is well-written; however, some improvements are needed before publication.

Major Revisions:

The authors should briefly describe endosome signaling, which plays a critical role in C-MET function.

The manuscript should introduce an important aspect related to C-MET–integrin cooperation.

The authors should incorporate more recent studies on C-MET function to ensure up-to-date references.

A table summarizing C-MET inhibitors along with relevant references should be included to clarify their activity.

The authors should also add a schematic representation illustrating the action of C-MET inhibitors to enhance the understanding of their effects.

Author Response

Response to Reviewers’ Comments

Reviewer Comment 1:

The authors should briefly describe endosome signaling, which plays a critical role in c-MET function.

Response:

Thank you for your insightful suggestion. In response, we have added a detailed description of endosomal signaling and its relevance to c-MET function in the revised manuscript (see page 4-5, lines 149-178). Specifically, we now discuss the role of ligand-bound c-MET trafficking to peripheral endosomes, the dependency of downstream signaling on the type of endosome involved, and how this affects ERK1/2 activation and cell migration. We also describe the retrograde trafficking of c-MET to the nucleus in response to reactive oxygen species and its implications for therapeutic resistance.

Reviewer Comment 2:

The manuscript should introduce an important aspect related to c-MET–integrin cooperation.

Response:

We appreciate this valuable comment. In the revised manuscript, we have added a new section discussing the bidirectional crosstalk between c-MET and integrin signaling (see page 5, lines 179–192). This includes both inside-out and outside-in signaling mechanisms, the role of integrins such as α5β1 in modulating c-MET phosphorylation, and the hypothesis regarding the involvement of integrin cytoplasmic domains. Relevant citations have been included to support this section.

Reviewer Comment 3:

The authors should incorporate more recent studies on c-MET function to ensure up-to-date references.

Response:

Thank you for your valuable suggestion. In response, we have revised and updated the manuscript to include recent studies that highlight new insights into c-MET function and therapeutic targeting. Specifically, we have added recent data from the Phase II LUMINOSITY trial (NCT03539536), which evaluated telisotuzumab vedotin (Teliso-V), an antibody–drug conjugate targeting c-MET, in nonsquamous, EGFR-wildtype NSCLC patients with c-MET overexpression. These findings not only reflect the current progress in c-MET–targeted therapy but also support the clinical relevance of c-MET expression as a predictive biomarker. (See page 14, line 577-586)

We have also updated the reference list accordingly to ensure the manuscript reflects the most recent advancements in the field.

Reviewer Comment 4:

A table summarizing c-MET inhibitors along with relevant references should be included to clarify their activity.

Response:

We fully agree with the reviewer’s recommendation. A summary table of representative c-MET inhibitors has been added to the revised manuscript (Table 2, page 15). This table includes each drug’s name, inhibitor type, general mechanism of action, and supporting references to enhance clarity and aid comparison across different therapeutic strategies.

Reviewer Comment 5:

The authors should also add a schematic representation illustrating the action of c-MET inhibitors to enhance the understanding of their effects.

Response:

Thank you for this suggestion. We have added a schematic figure (Figure 5, page 11) illustrating the general mechanisms of action of c-MET inhibitors, including both small molecule inhibitors and antibody-based therapies. This visual summary is intended to facilitate the reader’s understanding of how these therapies interfere with c-MET signaling at different levels.

Reviewer 2 Report

Comments and Suggestions for Authors

This is a fascinating review paper. I have a few comments,
1. One common mechanism for alternative splicing is SVA insertion, as described here (PMID: 37511314). The authors should describe the SVAs and other transposable elements in cancer. SVAs also induce gene/transcript fusions.

2. MET oncogene is also involved in osteosarcoma; I suggest the authors address this malignancy, too. Here is one paper with a comprehensive analysis (PMID: 38966281).

3. Does the MET gene have any viral interactions or virus-induced fusions?

4. I found a few typos in the manuscript.

Author Response

We sincerely thank the reviewer for your careful evaluation and constructive suggestions. The comments provided valuable insights that have helped us improve the clarity and completeness of our review. Below, we detail our responses and corresponding revisions made in the manuscript.

Comment 1:

One common mechanism for alternative splicing is SVA insertion, as described here (PMID: 37511314). The authors should describe the SVAs and other transposable elements in cancer. SVAs also induce gene/transcript fusions.

Response:

Thank you for this thoughtful comment. We recognize the emerging significance of SVA (SINE–VNTR–Alu) insertions and other transposable elements in cancer biology, particularly their potential to induce aberrant splicing and gene fusions. While we could not identify any current literature directly linking SVA insertions to MET gene alterations, we acknowledged the potential for such transposable elements to influence gene expression and splicing more broadly in cancer. This addition helps frame future research directions in the context of MET-related oncogenesis. The relevant discussion and citation (PMID: 37511314) have been incorporated into the revised manuscript (page 9, line 349-352).

Comment 2:

MET oncogene is also involved in osteosarcoma; I suggest the authors address this malignancy, too. Here is one paper with a comprehensive analysis (PMID: 38966281).

Response:

Thank you for highlighting this important point. We have now included a discussion on the role of c-MET in osteosarcoma based on findings by Zeng et al. (PMID: 38966281). The corresponding text has been added to page 10, line 368-373 of the revised manuscript.

Comment 3:

Does the MET gene have any viral interactions or virus-induced fusions?

Response:

Thank you for raising this important point. Currently, there is limited evidence directly linking the MET gene to viral interactions or virus-induced gene fusions. While certain oncogenic viruses (e.g., HPV, HBV, EBV) are known to induce genomic instability and gene fusions in human cancers, no virus-induced MET fusion has been definitively reported in the literature to date. Nonetheless, given the emerging recognition of viral elements in fusion gene formation, future studies may uncover such interactions involving MET.

Comment 4:

I found a few typos in the manuscript.

Response:

Thank you for pointing this out. We have carefully proofread the manuscript and corrected all identified typographical and grammatical errors throughout the text.

Reviewer 3 Report

Comments and Suggestions for Authors

Targeting c-MET Alterations in Cancer: A Review of Genetic 2 Drivers and Therapeutic Implications

 I have reviewed the manuscript and consider it to be well-written and comprehensive. The article presents a thorough review of current advances, addressing novel aspects of the field. This review highlights key MET alterations, such as gene amplification, gene fusions, and exon 14 skipping deletions, and examines their prevalence across various tumor types. It also discusses the clinical significance of c-MET as a therapeutic target and identifies gaps in knowledge that could inform the development of alternative treatment strategies.

In this review, the authors present an interesting section on drug treatment for genomic alterations in MET Currently, several therapeutic strategies have been developed to address these alterations: Crizotinib: ;  Capmatinib selective MET inhibitors: Type 2 and Type 3 inhibitors;Antibody-based therapies

The review provides a useful synthesis of recent knowledge and addresses a need within this field. It is well-structured and organized, with a clear division into the introduction, main body, and conclusions. Subheadings are effectively used to guide the reader and structure the content. The bibliographic selection is appropriate, including important and up-to-date references.

The topic is relevant and timely, and the manuscript is appropriate and offers valuable contributions to the understanding of emerging areas in this field. However, given the technical depth and specificity, it may be beneficial for the manuscript to be additionally reviewed by an oncologist with greater expertise in molecular oncology.

Author Response

We sincerely appreciate the reviewer’s thoughtful and positive evaluation of our manuscript. We are grateful for your recognition of the comprehensive and well-structured nature of our review, as well as your acknowledgement of its relevance and contributions to the understanding of MET alterations and therapeutic strategies.

We particularly appreciate your encouraging remarks on the organization, bibliographic selection, and clarity of the manuscript. Your feedback affirms the significance of our work in addressing current advances and knowledge gaps in the field.

Thank you once again for your time, thoughtful comments, and support.